# Physical Health, Media Use, Stress, and Mental Health in Pregnant Women during the COVID-19 Pandemic

**DOI:** 10.3390/diagnostics12051125

**Published:** 2022-05-01

**Authors:** Makarios Eleftheriades, Eleni Vousoura, Anna Eleftheriades, Panagiota Pervanidou, Iannis M. Zervas, George Chrousos, Nikolaos F. Vlahos, Alexandros Sotiriadis

**Affiliations:** 1Second Department of Obstetrics and Gynecology, School of Medicine, National and Kapodistrian University of Athens, Aretaieion Hospital, 11528 Athens, Greece; nfvlahos@gmail.com; 2Department of Psychology, National and Kapodistrian University of Athens, 15780 Athens, Greece; evousoura@psych.uoa.gr; 3First Department of Psychiatry, School of Medicine, National and Kapodistrian University of Athens, “Aiginiteion” Hospital, 11528 Athens, Greece; yanizervas@gmail.com; 4Postgraduate Programme in Fetal Maternal Medicine, Medical School, National and Kapodistrian University of Athens, 11527 Athens, Greece; annielefth-28@hotmail.com; 5Unit of Developmental and Behavioral Pediatrics, First Department of Pediatrics, National and Kapodistrian University of Athens, Aghia Sophia Children’s Hospital, 11527 Athens, Greece; nenyperva@gmail.com (P.P.); chrousos@gmail.com (G.C.); 6Second Department of Obstetrics and Gynecology, School of Medicine, Aristotle University of Thessaloniki, “Ippokrateion” Hospital, 54642 Thessaloniki, Greece; asotir@gmail.com

**Keywords:** pregnancy, COVID-19, maternal stress, maternal health, depressive symptoms

## Abstract

Background: The COVID-19 pandemic has led to significant changes in the care of pregnant women and their fetuses. Emerging data show elevated depression and anxiety symptoms among pregnant women. Aims: The purpose of this article is to investigate the psychological and behavioral impact of the COVID-19 pandemic on pregnant women in Greece during the first national lockdown. Methods: We used a cross-sectional, anonymous survey to collect data in two fetal medicine clinics in the largest urban centers of Greece during the months of April and May 2020. The questionnaire was largely based on the CoRonavIruS Health Impact Survey (CRISIS), and assessed sociodemographic characteristics, general health and obstetric data and COVID-19-related worries and life changes. Mood symptoms, substance use and lifestyle behaviors were assessed at two time points (3 months prior to the pandemic and the 2 weeks before taking the survey), while perceived stress was measured with the perceived stress scale (PSS-14). Results: A total of 308 pregnant women (*M*_age_ = 34.72), with a mean gestation of 21.19 weeks participated in the study. Over one-third of the women found COVID-19 restrictions stressful, and their highest COVID-19-related worry was having to be isolated from their baby. Mean PSS-14 score was 21.94, suggesting moderate stress. The strongest predictors of stress were physical and mental health status before COVID-19 and having experienced a stressful life event during their pregnancy. Compared to 3 months before the pandemic, women reported higher scores on mood symptoms (*p* < 0.001)*,* TV use (*p* = 0.01) and social media use (*p* = 0.031) in the last 2 weeks before taking the survey. Conclusion: Our study provides important preliminary evidence of the negative impact of the COVID-19 pandemic and the lockdown on pregnant women’s well-being and functioning.

## 1. Introduction

The novel coronavirus SARS CoV-2 disease (COVID-19) was declared a pandemic by the World Health Organization (WHO) on 11 March 2020. This unprecedented crisis has severely affected the delivery of medical care globally. Among the most vulnerable groups are pregnant women (PW), who face significant fear and worry over the impact of this novel virus on their pregnancy and fetal development and who must endure the many disruptions and changes in their antenatal care during the pandemic [1].

The impact of SARS-CoV-2 infection during gestation remains largely unclear. Published, well-documented cases of probable vertical transmission are scarce, suggesting that congenital infection is not common (about 2% of maternal infection) [2]. Physiological changes during pregnancy may increase the risk of developing severe illness in response to viral infections. However, it appears that PW are not more likely to acquire the infection, but there are data suggesting that the acute respiratory syndrome induced by SARS-CoV-2 could be more severe during pregnancy [3]. Moreover, PW may manifest a worse clinical course than nonpregnant females of similar age [4]. Nonetheless, these findings are to be taken seriously, as evidence suggests that acute respiratory virus infections, including coronaviruses and influenza, may lead to long-term neurodevelopmental and neuropsychiatric symptoms in the offspring [5,6]. Potential mechanisms may include a systemic inflammation overload with loss of placental integrity, fetal and maternal immune responses, and production of antineuronal antibodies, as well as a direct brain infection of the fetus [7].

To mitigate the risk of COVID-19 exposure and transmission in PW, changes in obstetric practice have been implemented globally. Maternity wards have enforced visitor restriction policies, and several hospitals prohibit or limit companions or escorts [8]. Routine perinatal care visits have been either reduced to a minimum or transitioned into a virtual mode of delivery [9]. The American College of Obstetricians and Gynecologists (ACOG, https://www.acog.org/ (accessed on 15 June 2020)) [10] and the Society for Maternal-Fetal Medicine (SMFM https://www.smfm.org/ (accessed on 15 June 2020)) have suggested modifications of traditional protocols for prenatal visits. Moreover, PW may have restricted access to public health systems, negatively affecting admittance by mental health services and decreased social support. There is limited information on the effects of these conditions on maternal and pregnancy outcomes; however, following lockdown measures, increased rates of stillbirth have been reported that could be associated with either disruptions in prenatal care services or with an increased incidence of home birth [11,12].

Emerging data from cross-sectional studies show elevated depression and anxiety symptoms among PW compared to pre-COVID-19 estimates [13,14,15]. Higher symptoms of depression and anxiety have also been associated with PW’s greater health concerns about themselves and their fetuses and more worries about the impact of the pandemic on prenatal care and social relationships and connectedness [16]. Data on the psychological impact of the COVID-19 pandemic echo epidemiological findings from past infectious outbreaks, including MERS-CoV, SARS, and Zika [17]. 

These findings are not to be taken lightly, given that pregnancy is an already vulnerable period for the development or recurrence of mental disorders. It is now well-known that the prevalence of mood and anxiety disorders in PW is higher than in the general population. According to ACOG, perinatal depression constitutes one of the most common medical complications of pregnancy, estimated to affect up to 16% of PW [18]. Antenatal anxiety is even higher; as reported in a recent meta-analysis, 20.7% of PW had at least one anxiety disorder, and 5.5% had at least two [19] anxiety symptoms. These disorders are generally more prevalent during the third trimester of pregnancy [20]. 

Stress and mood symptoms during pregnancy not only negatively affect the expectant mother’s wellbeing and pregnancy experience. They also increase the risk of postpartum depression, which has been associated with negative obstetric outcomes, and may have long-term consequences for the offspring [21,22]. The period of intrauterine life is among the most sensitive developmental windows, in which the effects of stress may be transmitted intergenerationally from mother to child. There is substantial evidence suggesting that exposure to excess stress during fetal life may affect several aspects of health and development of the offspring. Stress-related maternal, placental and fetal biological alterations and relative elevations in hormones, cytokines and neurotransmitters may affect the developing fetus in a process that has been named “fetal programming” [23,24]. Maternal stress is frequently associated with a chronic activation of the stress system, including the hypothalamic–pituitary–adrenal axis and the sympathetic nervous system [25]. Maternal stress may also be associated with stress-related unhealthy behaviors, such as overeating, smoking and poor adherence to self-care activities. Both biological and behavioral parameters may affect the fetus and are connected with adverse health outcomes. Indeed, there is evidence of short- and long-term adverse consequences in growth, neurodevelopment, and metabolic and cardiovascular health of the child [25,26]. 

In light of the negative sequelae of antenatal mood and anxiety disorders on fetal and infant development, ACOG recommends integration of screening for perinatal depression in the primary care of pregnant women. Obstetricians–gynecologists and other obstetric care providers should screen patients at least once during the perinatal period for depression and anxiety symptoms [27]. 

## 2. The Case of Greece and Aims of the Present Study 

The first COVID-19 case in Greece was reported on 26 February 2020. On 23 March, the country went on a generalized 6-week lockdown; retail stores were closed, and schools and academic institutions operated exclusively online. Only the health workforce and other essential workers, i.e., civil servants, cleaning staff, grocery store employees, and restaurant delivery staff, were allowed to be physically present at work. The number of positive cases during April and May was 1510, while the death rate was among the lowest globally (175 total deaths until May 2020). Several modifications in prenatal care services were implemented: prenatal protocols were revised, the number of in-person visits and the timing of visits were reduced, physical distancing was enforced, telehealth and communication via phone calls or emails were encouraged, while prenatal tests were grouped and booked for the same visit/day to minimize maternal physical contacts.

Very limited literature exists on the impact of the COVID-19 pandemic on the mental health and wellbeing of PW in Greece. Preliminary findings with 269 women attending routine antenatal care in a university clinic showed a sharp increase of anxiety levels in the first week of the lockdown followed by a subsequent decrease in the following weeks [28]. An important limitation of that study, however, was that it did not assess various important parameters, such as exposure to COVID-19, life changes as a result of the lockdown, and worries and anxiety about the impact on physical health.

Our study aims to address this critical gap by elucidating the psychological and behavioral impact of the COVID-19 pandemic on PW in Greece during the first national lockdown in the months of April–May. Specifically, our study aimed to examine: (1) changes over time in physical, mental health, media and substance use; (2) the unique contribution of stress and worry to the worsening of mood symptoms from 3 months before to the 2 weeks before completing the survey; and (3) the contribution of stress and worry to negative changes over time in physical health and media and substance use variables.

## 3. Materials and Methods

### 3.1. Sample

The study is a cross-sectional survey. Recruitment involved a consecutive sample of women attending an antenatal clinic, who provided an informed consent to participate in the study. Data were collected during the first lockdown, between April and May 2020. Eligible participants were women aged 18 years or above living in Greece, who were pregnant and attended their routine antenatal care at the time of data collection.

### 3.2. Procedure

The questionnaires were completed in hard copies in the waiting room of two different fetal medicine clinics, one in Athens and one in Thessaloniki, the two largest cities in Greece. These fetal medicine centers also provide services to referred patients from other adjacent areas.

Eligible participants provided their written consent to proceed to the completion of the questionnaire. Participation was anonymous and participants were informed through a written consent form that they could stop the questionnaire completion at any time.

The study protocol was approved by the Ethics Committee of the “University Research Institute of Maternal and Child Health and Precision Medicine” and complied fully with the Helsinki Declaration on human participant research.

### 3.3. Measures

#### 3.3.1. Translation Procedure

The questionnaire was largely based on the CoRonavIruS Health Impact Survey (CRISIS) [29], which was developed through a collaborative effort between the research teams at the National Institute of Mental Health Intramural Research Program Mood Spectrum Collaboration at the Child Mind Institute and the NYS Nathan S. Kline Institute for Psychiatric Research. The questionnaire was translated into Greek following the WHO guidelines for translation and adaptation of instruments [30]. Below, the different sections are described in detail.

#### 3.3.2. Participant Characteristics

Sociodemographic characteristics assessed included age, race/ethnicity, self-rated health, urbanicity, education, household size, health insurance coverage and receipt of government assistance. Health variables included self-rated physical and mental health rated poor to excellent and history of chronic medical problems.

A section on obstetric characteristics was added to the standard CRISIS tool, (http://www.crisissurvey.org/download/ (accessed on 1 July 2021): Greek) following consultation with fetal/maternal medicine experts. Variables included maternal weight and height, week of gestation, current pregnancy complications, obstetric history, number of pregnancies and miscarriages, number of children and stressful life events during pregnancy.

#### 3.3.3. COVID Worries in the Past 2 Weeks

COVID worries were assessed with the CRISIS tool. Participants reported how much they worried during the past 2 weeks about infection, friends and family being infected, and possible impacts on physical and mental health, as well as time spent reading or talking about COVID-19. A total of 16 items rated on a 5-point Likert scale (1 = not at all, 5 = extremely) were included. In the current study, the questionnaire had excellent internal consistency (Cronbach’s α = 0.925).

#### 3.3.4. Life Changes Due to COVID-19

Life changes were assessed with the CRISIS tool. Participants reported on life changes due to the pandemic in the past 2 weeks. Changes included social contacts, effects on family relationships, changes in living situation and stressors associated with these changes.

#### 3.3.5. Mood Symptoms

Mood symptoms were assessed with the CRISIS tool. Participants answered 10 items assessing mood/anxiety, both in the past 2 weeks and during the 3 months prior to the pandemic. Items were scored on a 5-point Likert scale (1 = not at all, 5 = very much); items 2 and 3 had to be reverse scored. The total score ranged from 10 to 5, with higher scores indicating worse mood symptoms. In the current study, the questionnaire had adequate internal consistency, both for 3 months prior to the pandemic, as well as in the past 2 weeks (Cronbach’s α = 0.725 and Cronbach’s α = 0.734, respectively).

#### 3.3.6. Substance Use

Substance use was assessed with the CRISIS tool. Participants rated the frequency of use of tobacco, alcohol, marijuana, and other substances during the past 2 weeks and during the 3 months prior to the pandemic.

#### 3.3.7. Lifestyle Behaviors

Lifestyle behaviors were assessed with the CRISIS tool. Participants reported the average weekday and weekend bedtime and sleep duration, frequency of exercise, time spent outdoors and length of media use per day for the past 2 weeks and the 3 months prior to the pandemic.

#### 3.3.8. Perceived Stress

Perceived stress was measured with a perceived stress scale [31], a 14-item self-report questionnaire that assesses the frequency of thoughts and feelings related to stressful events in the last month. Items are scored on a 5-point Likert scale ranging from 0 (never) to 4 (very often) and total scores range from 0 to 56, with higher scores denoting high perceived stress. The PSS scale is widely used and has been validated for use in many languages. PSS has been validated in Greece with good psychometric properties [32]. The internal consistency of the scale in the current study was good (Cronbach’s α = 0.82).

### 3.4. Statistical Analysis

Statistical analysis was performed using SPSS software (Version 23). Continuous variables are expressed as ranges and mean ± SD or median ± interquartile range, and categoric variables are expressed as absolute and relative frequencies. To identify correlates of PSS scores, we computed bivariate correlations with all demographic and obstetric variables. All significant correlations (*p* < 0.05) were then entered into a multiple regression model with PSS scores as the dependent continuous variable.

To examine changes over time in physical and mental health, media use and substance use variables (Aim 1), we compared the paired data 3 months before and during COVID-19, using the Wilcoxon signed-rank test for numerical variables and McNemar’s test for ordinal variables. To examine the unique contribution of stress and worry to the worsening of mood symptoms from 3 months before to the past 2 weeks of the lockdown (Aim 2), we performed a multiple linear regression analysis model with PSS and Worry scores as predictors and mood change scores as the dependent variable (a continuous variable calculated by subtracting total mood score at 2 weeks from total mood score at 3 months), controlling for gestational trimester and mental health before the pandemic. To examine the contribution of stress and worry to negative changes over time in physical health, media use and substance use variables (Aim 3), we performed separate logistic regression analysis for each domain (negative change vs. no change/positive change) with PSS and Worry scores as predictor variables, controlling for gestational trimester and mental health before the pandemic.

## 4. Results

### 4.1. Sample Characteristics

A total of 308 PW consented to participate in the study. Participants’ mean age was 34.72 years (range 22–48 years), with a mean gestation of 21.19 weeks (range 2–39 weeks). The majority lived in two major urban centers of Greece (*N* = 213, 74.2%) and had a university degree or higher (*N* = 232, 75.3%). The vast majority (98.1%) lived with their partner/spouse and had a mean of 0.62 children (range 0–6). The majority of the sample was employed (44.4%) or on a leave (35.4%), while 11.3% of the participants were laid off or unemployed and looking for a job. The sociodemographic characteristics of the sample are presented in Table 1.

The health and obstetric characteristics of the sample are presented in Table 2. Mean number of pregnancies (including current one) was 1.88 (SD = 1.07, range 1–8). History of miscarriage was reported by 24.5% of the sample. The majority of the sample reported very good or higher physical health (72.2%) and mental health (76.9%). A total of 119 women reported having one or more medical problems, with allergies (58.8) and asthma (16.0) being the two most common medical problems reported. A total of 72 women endorsed current pregnancy complications, with thyroid and gestational diabetes being the two most commonly experienced complications. Finally, 17.6% of women reported having experienced at least one stressful life event.

### 4.2. Health Status and Life Changes Due to COVID-19

At the time of the survey, only two participants reported having been diagnosed with COVID-19 (of which, one had not been formally tested), and one reported COVID-19 symptoms but had not received a formal diagnosis. Almost none of the participants had been in contact with someone diagnosed with COVID-19: two women reported being in contact with a confirmed case of COVID-19, and one with a possible case of COVID-19 (these three women are the ones who reported being infected with COVID; see above).

On average, women had conversations with 3.51 people outside of their household. The majority of them (67.6%) had spent 2 or fewer days outside of home, while 77.1% of the sample reported interacting less with contacts outside of home. For 37.4% of the sample, COVID-19 restrictions were moderately to extremely stressful. A total of 60 %of the sample expressed moderate to extreme concern about the stability of their living situation, and 59.5% expressed moderate to extreme difficulty with cancellation of important life events due to the restrictions (Figure 1A,B).

### 4.3. Worries Related to COVID-19

Women reported worrying moderately to extremely about family members or friends becoming infected with COVID-19 (*N* = 215, 69.7%), whether it is safe, if infected, to be hospitalized (*N* = 216, 70.3%), having to self-quarantine/be isolated from family members (*N* = 217, 71.0%), and having to be isolated from the baby (*N* = 238, 77.6%). Women reported worrying less about becoming themselves infected by COVID-19 (60% reported moderate to extreme worry) and having to give birth by C-section (50.2% reported moderate to extreme worry; Table 3).

### 4.4. Perceived Stress

The mean score on the PSS-14 at 2 weeks was 21.94 (SD = 0.41), which is suggestive of moderate stress. PSS-14 total scores significantly correlated with all worry items (Spearman rho values ranging from 0.15 to 0.31, *p*-values ranging between 0.01 and 0.001) except the item “worry that I may have to give birth via C-section” (Table 3). PSS total scores were significantly correlated with gestational trimester (*r* = −0.164, *p* = 0.005), physical and mental health before COVID-19 (*r* = −0.181, *p* = 0.001 and *r* = −0.246, *p* < 0.001, respectively) and having experienced a stressful life event (*r* = 0.189, *p* = 0.001).

The results of the multiple regression analysis with trimester, physical and mental health before COVID-19 and experiencing a stressful life event as predictor variables and PSS-14 score in the past two weeks as the outcome variable showed that they accounted for 17% of the variance of PSS (*F* = 11.67, *p* <.001). Significant predictors for the increase in depression symptoms were mental health (β = −0.17, *t* = −2.99, *p* = 0.003), stressful life event (β = 0.130, *t* = 2.411, *p* = 0.02) and worries about COVID-19 (β = 0.26, *t* = 4.86, *p* < 0.001).

### 4.5. Differences in Mental Health, Physical Health, Substance Use, and Media Use before and during the Pandemic

On average, participants reported higher scores on mood symptoms in the last 2 weeks (*Mdn* = 25.0) compared to the past 3 months (*Mdn* = 22.0). A Wilcoxon signed-rank test indicated that the difference was statistically significant, *T* = 8490.5, z = −7896, *p* < 0.001. Compared to 3 months before, significantly more participants slept 8 h or more during the weekend in the past 2 weeks (*p* < 0.001), while no significant change in sleep hours was observed during the weekend. No significant changes were found in physical activity and outdoor activity. TV and social media use were significantly increased (*p* = 0.01 and *p* = 0.031, respectively). Substance use (alcohol, vaping, smoking, and sleeping pill use) were all significantly decreased (see Table 4).

### 4.6. Predictors of Differences in Mental Health, Physical Health, and Media Use before and during the Pandemic

Table 5 shows the unique contribution of PSS score and COVID-19 worries on changes in mood symptoms from 3 months before to the past two weeks of the lockdown. The multiple linear regression model was significant [*F*(4296) = 8.55, *p* < 0.001], accounting for 10% of the variance. Controlling for gestational trimester and mental health before the pandemic, both PSS (β = −0.22, *t* = 3.69, *p* < 0.001) and worry (β = −0.12, *t* = 2.00, *p* = 0.046) scores were significantly associated with worsening of mood symptoms at 2 weeks.

Table 6 shows the unique contribution of PSS and COVID-19 worries on the changes in physical health and media use during COVID-19. PSS significantly increased the odds of a negative change in sleep duration during the weekdays and daily TV use, while worries related to COVID-19 decreased the odds of a negative change in TV use.

## 5. Discussion

Our study provides important information regarding the impact of the COVID-19 pandemic and the lockdown on life changes, maternal worries and mental and physical health among PW. Thus, the COVID-19 pandemic undoubtedly brought many changes in the lives of PW. The women of our cohort were mostly concerned about the lack of stability in their living situations, the financial strains of the pandemic on their family and having missed social life events due to the imposed restrictions. They also worried about having access to maternal health care with a minimal exposure risk during the pandemic. In line with previous findings [33,34], PW worried less about their own health and more about the risk of infection during hospital visits and disruptions in their prenatal care and delivery caused by modifications of hospital visit protocols.

Our findings further suggest that PW experienced moderate stress during the months of April–May 2020, corroborating existing findings showing that the numerous restrictive measures imposed by the government and the consequent social distancing were stressful to expecting mothers [14,35]. Several factors predicted greater vulnerability to perceived stress. Women experienced higher levels of stress if they reported poor mental health before the pandemic and that they had experienced stressful life events during their pregnancy, both of which are well-established risk factors for antenatal depression and anxiety disorders [36]. Additionally, PW who worried about COVID-19 were more likely to report higher levels of stress. This finding is in line with previous research showing that higher symptoms of depression and anxiety were associated with greater concerns about threats of COVID-19 to the life of the mother and baby and concerns about not getting the necessary prenatal care, experiencing relationship strain and feeling social isolation due to the COVID-19 pandemic [16].

Interestingly, none of the obstetric characteristics of the sample was associated with perceived stress. Other studies have reported a relation between high-risk pregnancy and perceived stress. For instance, in a cross-sectional, case-control study, Sinaci et al. [37] found that high-risk pregnancy patients were not only at higher risk of becoming infected by SARS-CoV-2, but also experienced higher anxiety symptoms in the context of the pandemic. It is possible that the discrepancy with our results may be due to the relatively small percentage of women with high-risk pregnancies in our sample.

Our study also documented significant changes in the mental and physical health of PW since the onset of the pandemic. First, in line with recently published reports from all over the world on the mental health impact of the pandemic, women in our study reported a significant increase in mood and anxiety symptoms compared to 3 months before the pandemic outbreak. Wu et al. reported a significant increase in the prevalence of depression and anxiety symptoms, as well as in thoughts of self-harm, among PW after the official announcement of the COVID-19 outbreak by the Chinese government [13]. Lebel et al. [16] in their study with 1987 PW in Canada surveyed in April 2020, found that 37% of PW reported clinically relevant symptoms of depression, while 57% reported clinically relevant symptoms of anxiety, both estimates being substantially higher than findings from previous community pregnancy cohorts with similar demographic characteristics. Furthermore, in Turkey, a cross-sectional study conducted on 403 PW using a web-based survey confirmed increased prevalence of anxiety and depression symptoms during the pandemic [38]. Another study from Turkey, involving 260 PW, showed elevated scores in depression and anxiety screening instruments, with 35% of participants presenting symptoms of depression at a clinical level [15]. An Italian cross-sectional survey study conducted from 15 March to 1 April 2020, involving 100 PW, showed that more than half [53%] of the participants rated the psychological impact of COVID-19 as severe, while two-thirds were more anxious than normal [14].

We also found that, controlling for mental health status before the pandemic, worsening of mood and anxiety symptoms was predicted by both perceived stress and worry about COVID-19. Similar to our findings, in a cross-sectional study with 2740 PW in the US during 3–24 April 2020, Moyer et al. [35] found high levels of pregnancy-related anxiety; those who feared becoming infected had greater changes in perceived pregnancy-related anxiety. Additionally, other significant predictors of changes in pregnancy-related anxiety scores were modification of prenatal visits, changing birth plans, fear of food running out, increased tension/conflict in the home and living or working in an area endemic to COVID-19 disease. The abovementioned factors were significant even after adjustments for age, education or previous history of depression and anxiety.

During the lockdown, TV and social media use significantly increased. Perceived stress emerged as a predictor of increased TV use. As both traditional news and social media have been criticized as contributory to the phenomenon of “information pollution”, which in turn may trigger uncertainty and stress in consumers, management of media use appears crucial to safeguarding PW wellbeing. Interestingly, sleep duration during the weekdays improved, as more women reported getting eight or more hours of sleep during weekdays compared to 3 months before the pandemic. However, those who did report worsening of sleep during the weekdays were more likely to have experienced higher levels of stress. Other studies in the general population have shown that adults experienced a number of sleep disturbances during the initial stages of the COVID-19 pandemic [39]. Furthermore, analysis of an international sample of tweets related to pregnancy and mental health during the first wave of COVID-19, from March to June 2020 revealed, among other concerns, high prevalence of sleeping problems [40]. Further investigation of the putative sleep changes is warranted, given how crucial sleep hygiene is for the health and wellbeing of PW.

Finally, in our study, substance use decreased among PW. While studies in the general population have reported a rise of substance use disorders since the beginning of the COVID-19 outbreak [41] other studies specific for the pregnant population have shown that avoiding tobacco, alcohol and other substances, and engaging in a healthy lifestyle may be effective coping strategies to manage the stress of the situation [34].

## 6. Conclusions

COVID-19 resulted in significant disruptions in the antenatal care of PW, leading to moderate stress and worries over the health of their fetus. Our study provides evidence of an increase in mood and anxiety symptoms compared to 3 months before the pandemic outbreak, as well as increases in TV and social media use, all of which were associated with greater stress. Some unexpected positive changes were also observed, including increased sleep duration and a decrease in substance use. These results illustrate the importance of managing stress and worries among PW during this period of heightened uncertainty. Stress management interventions should be implemented in prenatal care during the pandemic. As the number of health care visits are modified to a minimum, it is crucial that obstetricians–gynecologists and other obstetric care providers screen their patients for mental health problems and that maternal mental health services are effectively integrated in the existing health systems.

### Study Limitation

The cross-sectional design of the study renders any causal inferences impossible. The survey was conducted at the early stages of the pandemic, during which both confirmed cases and number of deaths were low in Greece. Our findings may not be representative of PW’s mental status in more recent times. As more people are getting sick, mental exhaustion is building up due to the imposed social restrictions, and the negative economic consequences are worsening. In addition, data collected on mental and physical health three months before COVID-19 were based on retrospective recall and are, thus, subject to recall bias. Future studies should implement a prospective longitudinal design to overcome the inherent limitations of cross-sectional research. Contrary to most studies conducted during the pandemic, our study implemented a face-to-face data collection strategy; while this approach has the advantage of including a representative sample of women attending routine antenatal care, high-risk PW may be less likely to enroll in such a study. Moreover, the sample was primarily urban, college-educated, and of Greek nationality only, which may limit generalizability of our findings to low-income urban or rural groups and ethnic minorities. Furthermore, our questionnaire did not include any questions regarding the eating habits of pregnant individuals, which may have significantly changed during the lockdown. Finally, our study did not investigate some important variables that could be related to mood and anxiety disorders, particularly in the context of the pandemic, such as the quality of the relationship with the partner, potential intimate partner violence [9] and availability of social support [40]^.^

## Figures and Tables

**Figure 1 diagnostics-12-01125-f001:**
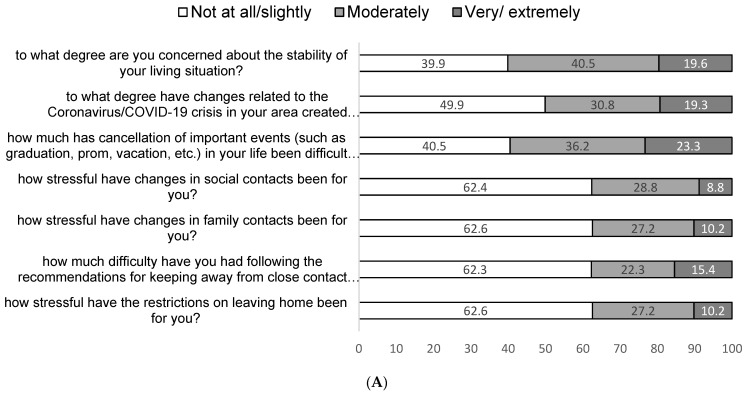
(**A**,**B**) Life changes during the COVID-19 pandemic (*N* = 308).

**Table 1 diagnostics-12-01125-t001:** Sociodemographic characteristics of the sample (*N* = 308).

Variables	Participants, *N* (%)
Age ^a^	22–48 (34.72 ± 4.64)
Children (not including current pregnancy) ^b^	0–6 (0.5 ± 1.0)
Greek nationality	303 (98.4)
Region	
*Athens/Thessaloniki*	213 (74.2)
*Other*	74 (25.8)
Educational Level	
*High school*	35 (11.3)
*Vocational training*	41 (13.3)
*University degree*	135 (43.8)
*Graduate degree*	97 (31.5)
Living with partner (yes)	302 (98.1)
Living with parents (yes)	16 (5.2)
Relationship Status	
*Married*	271 (90.0)
*Partnered*	25 (8.3)
*Single/Separated*	5 (1.7)
Employment status	
*Employed*	135 (44.4)
*Maternity leave*	107 (35.4)
*Unemployed*	34 (11.3)
*Retired*	1 (0.3)
*Homemaker*	22 (7.3)
*Disability*	4 (1.3)
No. of household members ^b^	0–7 (2.0 ± 1.0)
No. of rooms ^b^	2–10 (3.00 ± 2.0)
Essential worker living in the household (yes)	185 (71.7)
On state funding (yes)	48 (15.6)

Note: ^a^ range (mean ± standard deviation); ^b^ range (median ± interquartile range).

**Table 2 diagnostics-12-01125-t002:** Health status and obstetric information (*N* = 308).

Variables	Participants, *N* (%)
BMI ^a^	16.41–41.50 (25.63 ± 4.71)
Gestational week ^a^	2–39 (21.19 ± 8.24)
*First trimester*	74 (24.6)
*Second trimester*	154 (51.2)
*Third Trimester*	73 (24.3)
Number of pregnancies ^b^	1–8 (2.00 ± 1)
History of miscarriage (yes)	74 (24.5)
Number of miscarriages ^b^	0–6 (0 ± 0)
Physical health (very good/excellent)	221 (72.2)
Mental health (very good/excellent)	237 (76.9)
Medical problems	119 (38.6)
Pregnancy complications (yes)	72 (23.4)
*Thyroid*	42 (13.6)
*Gestational Diabetes*	24 (7.8)
*Hypertension*	5 (1.6)
*Preeclampsia*	3 (1.0)
*Other*	6 (8.3)
Stressful Life Events during pregnancy (yes)	54 (17.6)

Note: BMI = Body Mass Index; ^a^ range (mean ± SD); ^b^ range (median ± interquartile range).

**Table 3 diagnostics-12-01125-t003:** Worries related to COVID-19 and bivariate correlations between worries and PSS ^a^ for the total sample (*N* = 308).

Worries	*N* (%)	PSS Total Score
Becoming infected		0.27 ***
Not at all—little	123 (40)	
Moderate	111 (36.2)	
A lot—extremely	73 (23.8)	
Family members or friends becoming infected		0.23 ***
Not at all—little	93 (30.3)	
Moderate	95 (30.8)	
A lot—extremely	120 (38.9)	
Being at high risk for becoming infected by COVID-19		0.22 ***
Not at all—little	99 (32.1)	
Moderate	107 (34.7)	
A lot—extremely	102 (33.1)	
Being at risk for a miscarriage or other obstetric complication		0.22 ***
Not at all—little	123 (40.2)	
Moderate	73 (23.9)	
A lot—extremely	110 (36.0)	
Fetal infection leading to genetic disorders		0.21 ***
Not at all—little	128 (42.0)	
Moderate	72 (23.6)	
A lot—extremely	105 (34.4)	
Fetal infection leading to health disorders		0.24 ***
Not at all—little	118 (38.8)	
Moderate	69 (22.7)	
A lot—extremely	117 (38.5)	
Whether it is safe to attend OBGYN appointments/give birth in hospital		0.25 ***
Not at all—little	121 (39.4)	
Moderate	97 (31.6)	
A lot—extremely	89 (29.0)	
Whether it is safe, if infected, to be hospitalized		0.23 ***
Not at all—little	91 (29.7)	
Moderate	83 (27.0)	
A lot—extremely	133 (43.3)	
Having to self-quarantine/isolate from other family members		0.23 ***
Not at all—little	89 (29.1)	
Moderate	77 (25.2)	
A lot—extremely	140 (45.8)	
COVID-19 infection causing birth complications		0.18 **
Not at all—little	108 (35.1)	
Moderate	65 (21.1)	
A lot—extremely	135 (43.9)	
Having to give birth via C-section		0.09
Not at all—little	92 (49.9)	
Moderate	63 (20.7)	
A lot—extremely	90 (29.5)	
Not being able to breastfeed		0.15 ***
Not at all—little	102 (33.2)	
Moderate	72 (23.5)	
A lot—extremely	133 (43.3)	
Isolating from baby after birth		0.26 ***
Not at all—little	69 (22.4)	
Moderate	57 (18.6)	
A lot—extremely	181 (59.0)	
Physical health being affected by COVID-19		0.15 ***
Not at all—little	93 (30.2)	
Moderate	93 (30.2)	
A lot—extremely	120 (39.2)	
Mental health being affected by COVID-19		0.23 ***
Not at all—little	115 (37.3)	
Moderate	93 (30.2)	
A lot—extremely	100 (32.5)	
Worry Total Score		0.31 ***

Notes: ^a^ Spearman rank correlation coefficient; ** *p* < 0.01 ****p* < 0.001.

**Table 4 diagnostics-12-01125-t004:** Differences in current physical health, substance use, media use, and mental health compared to 3 months prior to COVID-19 (*N* = 308).

	3 Months Prior	Past 2 Weeks	*p*-Value
	*N* (%)	*N* (%)	
Physical Health			
<8 h of sleep week	212 (68.8)	173 (56.2)	*p* < 0.001 ^a^
<8 h of sleep weekend	180 (58.4)	177 (57.5)	*p* > 0.05 ^a^
<5 days/week physical activity	278 (90.3)	272 (89.5)	*p* > 0.05 ^a^
<5 days going out	244 (81.9)	261 (85.9)	*p* > 0.05 ^a^
Mood symptoms			
Total Mood Symptoms (Median)	22.0	25.0	*p* < 0.001 ^b^
Media use			
>1 h of TV per day	178 (57.8)	203 (66.1)	*p* = 0.01 ^a^
>1 h of social media per day	129 (42.2)	151 (49.0)	*p* = 0.031 ^a^
>1 h of gaming per day	30 (9.9)	35 (11.4)	*p* > 0.05 ^a^
Substance use ^†^			
Alcohol	81 (36.2)	13 (5.7)	*p* < 0.001 ^a^
Vaping	21 (9.3)	4 (1.8)	*p* < 0.001 ^a^
Smoking	28 (12.14	6 (2.7)	*p* < 0.001 ^a^
Marijuana/cannabis	7 (3.1)	2 (0.9)	*p* > 0.05 ^a^
Opioids	1 (0.4)	0 (0.0)	*p* > 0.05 ^a^
Other drugs	2 (0.7)	1 (0.4)	*p* > 0.05 ^a^
Sleeping pills	44 (19.4)	0 (0.0)	*p* < 0.001 ^a^

Notes: ^†^
*N* = 227 (subsample of women of second and third trimester); ^a^ McNemar’s test; ^b^ Wilcoxon signed-rank test.

**Table 5 diagnostics-12-01125-t005:** Multiple linear regression model estimating the effect of PSS and COVID-19 worries on changes in mood symptoms (*N* = 308).

	Unstandardized Coefficients		95% CI	Standardized Coefficients		
Variable	B	SE	LL UL	β	*T*	*p*
Constant	−11.62	2.61	[−16.75, −6.49]		−4.46	0.000
Gestational trimester	0.36	0.46	[−0.53, 1.26]	0.04	0.80	0.425
Prepandemic mental health status	1.84	0.44	[−0.98, 2.71]	0.24	4.19	0.000
COVID-19 Worries	0.05	0.03	[0.00, 0.10]	0.12	2.00	0.046
PSS total score	0.18	0.05	[0.08, 0.27]	0.22	3.69	0.000

Note: *R*^2^ = 0.104.

**Table 6 diagnostics-12-01125-t006:** Association between stress and worries and changes in functioning compared to 3 months prior to pandemic (*N* = 308).

	No Change/Positive Change	Negative Change	Adjusted OR for Stress Score	Adjusted OR for Worries Score
Hours of sleep per week	265 (88.3)	35 (11.7)	1.06 (1.00–1.11) *	1.02 (0.52–1.32)
Hours of TV per day	198(64.5)	109 (35.5)	1.06 (1.02–1.10) ***	0.97 (0.95–0.99) **
Hours of social media per day	225 (73.5)	81 (26.5)	1.02 (0.98–1.06)	0.98 (0.96–1.00)

Notes: Binary logistic regression models controlled for gestational trimester and mental health pre-COVID; * *p* < 0.05 ** *p* < 0.01 *** *p* < 0.005.

## Data Availability

All data generated or analyzed during this study are presented in this paper. All other data are available from the corresponding author on reasonable request.

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
