# Peer review of "Physical Health, Media Use, Stress, and Mental Health in Pregnant Women during the COVID-19 Pandemic"

_diagnostics, 2022, doi:10.3390/diagnostics12051125_

Round 1
Reviewer 1 Report
The work presents an interesting aspect of the pandemic due to COVID19
As could be expected, especially in this delicate moment of life, the impact of the pandemic is important.
- It would be interesting a stratification according to the trimester of pregnancy, as there are variations (independent of the pandemic)
- You have no information on the practice of physical/sporting activity? Both in the course of the test and previously?
- You do not have data on eating habits and whether these have changed according to the changes recorded?
Reviewer 2 Report
I appreciate the opportunity to review the manuscript titled “Physical health, media use, stress, and mental health in pregnant women during the COVID-19 pandemic” that was submitted to the Diagnostics. The paper aims to investigate the impacts of COVID-19 lockdown on pregnant women’s mental health and some behaviors in Greece.
This paper is well written. The study addresses an important issue about pregnant women’s health during the COVID-19 pandemic. The overall methodology and study design are appropriate and current with measuring a broad range of variables and with before and after comparisons. The results and findings are clear and well-reasoned. Also, important study limitations are acknowledged.
Here are my comments:
The study investigated a broad range of variables, and I could see the statistical analysis and the results sections followed a logical order to demonstrate the study findings. Adding specific research objectives (such as, 1. to investigate the changes in physical health, mental health, media use, and substance use before and after…; 2. ; 3. ) could be more helpful and could provide a guide for readers to follow this logical flow, especially for the methods and results sections and the tables/figures.
All tables and Figures: please add (N= ) after the tables/figures’ captions/headings, so readers could have an idea about the total sample size.
Table 1
Essential worker living in the household: Is this a question with yes and no answers? May add (yes) after this variable to clarify if I understood correctly.
Table 2
Number of miscarriages 0-6 (0,0): What does this mean: (0,0)?
What does this refer to: * after Number of pregnancies and Number of miscarriages?
Suggest to add the full name of BMI at the bottom of the table.
Table 3
I understand Figure 2 displayed the worry scale results, however, after reviewing Figure 2, I could not have an accurate vision of the results, such as the percentages of people with moderate to extreme worry scores on each item. Here is my suggestion for authors to consider and decide, adding worry scale summary results to Table 3 and deleting Figure 2. Then, together with the 4.3 paragraph, the worry scale results could be clearly displayed.
Table 6
There are 3 superscripts a, what does it mean?
In Figure 1(A), it would be great if some summary results could be added to the figure, such as percentages of people with moderate to extreme scores for each item.
In Figure 2 (B), it would be also helpful if percentages of each item (a lot more, a little more…) could be added to the figure.
Round 2
Reviewer 1 Report
Even the authors could not act most of the improvements I suggested I think is suitable for pubblication